# Subarachnoid Haemorrhage—Incidence of Hospitalization, Management and Case Fatality Rate—In the Silesian Province, Poland, in the Years 2009–2019

**DOI:** 10.3390/jcm11144242

**Published:** 2022-07-21

**Authors:** Beata Łabuz-Roszak, Michał Skrzypek, Anna Starostka-Tatar, Anetta Lasek-Bal, Mariusz Gąsior, Marek Gierlotka

**Affiliations:** 1Department of Neurology, Institute of Medical Sciences, University of Opole, 45-040 Opole, Poland; 2Department of Biostatistics, Faculty of Health Sciences in Bytom, Medical University of Silesia, 40-055 Katowice, Poland; mskrzypek@sum.edu.pl; 3Department of Health Care, University of Applied Sciences in Ruda Slaska, 41-712 Ruda Slaska, Poland; annastarostka@wp.pl; 4Department of Neurology, Faculty of Health Sciences in Katowice, Medical University of Silesia, 40-055 Katowice, Poland; alasek@gcm.pl; 5Department of Cardiology, Faculty of Medical Sciences in Zabrze, Medical University of Silesia, 40-055 Katowice, Poland; m.gasior@op.pl; 6Department of Cardiology, Institute of Medical Sciences, University of Opole, 45-040 Opole, Poland; marek.gierlotka@uni.opole.pl

**Keywords:** subarachnoid haemorrhage, epidemiology, incidence, mortality

## Abstract

Little is known about the epidemiology of subarachnoid haemorrhage (SAH) in Poland, and until now no such research has been conducted for Silesia, which is the second largest province with circa 4.5 million inhabitants. Therefore, the current study was done to assess the data on SAH in the Silesian Province, Poland. The study was based on the data obtained from the administrative databases of the only public health insurer in Poland (the National Health Fund, NHF) from 2009 to 2019. The SAH cases were selected based on primary diagnosis coded in ICD-10 as I60. The total number of SAH cases was 2014 (41.8% men, 58.2% women). The number of SAH hospitalizations decreased from 199 in 2009 to 166 cases in 2019; *p* < 0.05. The median age increased from 58 in 2009 to 62 years in 2019; *p* < 0.001. Endovascular treatment or clipping of the aneurysm was performed in 866 cases (43%). An increase in in-hospital mortality was observed from 31% in 2009 to 38% in 2019 (*p* = 0.013). Despite the number of stroke units increasing, in-hospital mortality in SAH patients is high, and the number of vascular interventions seems insufficient. Better organization for care of SAH patients is needed in Poland.

## 1. Introduction

Subarachnoid haemorrhage (SAH) is a type of haemorrhagic stroke characterized by high morbidity and mortality [1,2]. After traumatic and iatrogenic causes, the next cause of SAH is ruptured cerebral aneurysm [2]. Arteriovenous malformation, cavernous haemangioma, cerebral arteriovenous fistula, and venous thrombosis could also be the cause of blood extravasation [3]. Other less common causes are tumour bleeding, vasculitis, encephalitis, cerebral amyloid angiopathy, postpartum eclampsia and reversible cerebral vasoconstriction syndrome (RCVS) [3,4]. SAH can cause various consequences for the individual, ranging from none to severe sequelae, such as death, physical disability, cognitive impairment, depression and anxiety, which affect the quality of life [2,3,4,5,6,7].

Incidence of SAH varies from 2.3 to 21.5 per 100,000 persons per year, depending on age, gender, region and country [5]. According to a recent systematic review and meta-analysis, global SAH incidence declined from 10.2 per 100,000 person-years in 1980 to 6.1 in 2010 (incidence of SAH decreased by about 40% in Europe, 46% in Asia, and 14% in North America but increased by almost 60% in Japan) [5].

Determining the incidence and risk factors of SAH is crucial to undertake preventive actions. However, little is known about the epidemiology and trends of SAH in Poland over years [1,8,9,10], and until now no epidemiological research on SAH has been conducted for the Silesian Province, which is the second largest province with circa 4.5 million inhabitants (12% of Poland’s population).

Therefore, the aim of the study was to evaluate the epidemiological situation related to SAH, i.e., the number of hospitalizations, management, and in-hospital and post-discharge mortality, in the Silesian Province, Poland, in the years 2009–2019.

## 2. Materials and Methods

The study was based on data obtained from the administrative databases of the only public health insurer in Poland (the National Health Fund, NHF) from 2009 to 2019.

For this analysis, SAH cases were selected based on primary diagnosis classified in the International Classification of Diseases (ICD-10) as I60 during hospitalization in any ward of any hospital in the Silesia Province. Additionally, patients who died in the emergency department with a primary diagnosis of SAH were included. Patients with comorbid diagnosis of head trauma (classified in ICD-10 as S00-S09) were excluded from the study. Patients under 18 years of age on admission and those who were not residents of the Silesian Province were also excluded. As the SAH is an emergency condition, it was assumed that details of practically all hospitalized cases, and also uninsured patients, were included in the database. Hypertension was defined as prior hospitalization with the following ICD-10 diagnoses: I10-I15. Then, for each year during the period of 2009–2019, index hospitalization was identified and analysed. Index hospitalization was defined as continued in-patient stay, through all the possible transfers between wards or hospital for any reasons, until the patients were discharged or died. Data on post-discharge deaths were obtained from the official government databases held by the NHF.

No approval of the bioethics committee was required, as the study was not a medical experiment and was an analysis of blind administrative data without any influence of diagnostic or therapeutic process.

### Statistical Analysis

Categorical variables were presented as values and percentages and the χ^2^ test was used for comparison between the groups. Continuous variables were expressed as mean and standard deviation or median with interquartile range. The normality assumption was tested using the Shapiro-Wilk test. To assess temporal trends, the Jonckheere–Terpstra test and the Cochran–Armitage test were used for continuous and categorical data, respectively. The Holm–Bonferroni correction for multiple comparisons was used where appropriate. We assumed the gamma distribution to calculate 95% confidence intervals for hospitalization rates. All analyses were performed by means of SAS 9.4 (SAS Institute Inc.; Cary, NC, USA). Statistical significance level was set to α = 0.05.

## 3. Results

The total number of SAH cases in the Silesian NHF databases for 2009–2019 was 2014, 842 male (41.8%) and 1172 female (58.2%).

The number of hospitalizations due to SAH decreased from 199 in 2009 (5.36 per 100,000 adult inhabitants of the Silesian Province) to 166 cases in 2019 (4.53 per 100,000); *p* = 0.0005. The ratio of male to female patients was stable over the years (*p* = 0.476) (Table 1).

The mean age of patients with diagnosed SAH was 62.2 years and it increased from 59.5 in 2009 to 61.6 in 2019; *p* < 0.001 (Table 2). The percentage of younger patients (<55 years) decreased (*p* = 0.006) while the percentage of elderly patients (>75 years) increased over the years (*p* < 0.001).

Previous history of hypertension (I10-I15) was present in 638 cases (31.7%), more in 2019 (39%) than in 2009 (21%) (Figure 1).

A total of 58.2% of SAH patients (*n* = 1172) were hospitalized in the neurosurgery ward, 28.3% (*n* = 570) in the intensive care unit and 69.7% (*n* = 1404) in the neurology ward or stroke unit. Interventions (coiling/other endovascular treatment or clipping of the ruptured aneurysm) were performed in 866 cases (43%) (Table 3). The remaining patients were treated conservatively (*n* = 1148; 57%). Patients who underwent intervention were younger (56.8 ± 7 vs. 67.1 ± 11 years), more often female (63.4% vs. 54.3%) and less frequently diagnosed with hypertension (21% vs. 40%) than those treated conservatively. Although the percentage of patients hospitalized in the neurosurgery ward gradually decreased over the years (Figure 2), the number of interventions remained similar (Table 3). The number of SAH patients who underwent endovascular treatment of the ruptured aneurysm increased over the years: in 2009 it constituted 28.7% of all interventions and in 2019 it was 58.3% (*p* < 0.001) (Table 3). Treatment of ruptured aneurysm was performed in 46.8% of women (*n* = 549) and in 37.6% of men with SAH (*n* = 317) (*p* < 0.0001). Younger patients were treated statistically significantly more often than elderly patients (60.4% of patients <55 years, 45.3% of patients between 55 and 75 years, and 14% of patients >75 years; *p* < 0.0001).

Median hospitalization time (from first admission to final discharge) decreased significantly over the years (in 2009 it was 16 days, in 2019 it was 14 days; Jonckheere–Terpstra trend test: *p* = 0.030) (Figure 3).

The overall in-hospital mortality in SAH patients in 2009–2019 was 36.5% (*n* = 735), and an increasing trend was observed (Cochran–Armitage trend test: *p* = 0.013) (Figure 4). In-hospital mortality was higher among patients treated conservatively (*n* = 551; 48%) than in patients who underwent intervention (*n* = 184; 21.2%) (*p* < 0.001; χ^2^ test) (Figure 5).

Among patients who were discharged from the hospital (*n* = 1279), 174 (13.6%) died within twelve months from the onset of the disease, including 110 patients (18.4%) treated conservatively and 64 patients (9.4%) who underwent intervention (*p* < 0.001; χ^2^ test). The overall post-discharge 12-month mortality in SAH patients and the post-discharge 12-month mortality in patients treated conservatively and in patients who underwent intervention are presented in Figure 6 and Figure 7.

In total, 909 post-SAH patients (45.1%) died within a year, with 661 (72.7%) patients treated conservatively and 248 (27.3%) after intervention (*p* < 0.001; χ^2^ test). 

Both in-hospital mortality and post-discharge 12-month mortality were lower in younger patients (*p* < 0.0001) and those who underwent aneurysm interventions (*p* < 0.0001) but were not gender specific. Overall combined mortality (in-hospital mortality and post-discharge 12-month mortality), as well as combined mortality in patients treated conservatively and in patients who underwent intervention, are presented in Figure 8 and Figure 9.

In 81 cases (4%), recurrent SAH was observed, in 39 subjects (1.9%) within 6 months, in 14 subjects between 6 and 12 months (0.7%), and in 28 subjects (1.4%) over 12 months.

## 4. Discussion

The above analysis showed a downward trend in the hospitalized incidence of subarachnoid haemorrhage in the Silesian Province in the years 2009–2019. The hospitalized incidence rate was 5.36/100,000 and 4.53/100,000 in 2009 and 2019, respectively. Similar trends were observed worldwide. The 2019 summary of global reports on the epidemiology of SAH estimated an annual decrease in the incidence by 1.7% in 1955–2014. The crude incidence rate was 6.1/100,000 person-years in 2010 compared to 10.2/100,000 person-years in 1980 on a global scale. Looking at the data continent-wise, in Europe this decrease has also amounted to 1.7% annually since 1972, in Asia the decrease has been 2.0% since 1977, in North America 0.7% since 1955, in Australia and New Zealand 0.8% since 1982, and in South and Central America the incidence has increased by 1.4% since 1998. The data from Africa are insufficient to determine the change in the incidence of SAH [5]. In reports that date before 2005, the countries of Central and South America showed a lower incidence of SAH than the world average [7]. Among European countries, many publications mention Finland as a population in which the incidence of SAH has not decreased. However, new reports show that the continued high incidence in this country is the result of analysing only a part of patient groups and not the entire population [11]. Another country with high incidence is Japan, where the increase in incidence since 1977 has been 1.6% [5].

Patient age, gender and underlying diseases are also important in the epidemiology of SAH. The above analysis showed that the mean age of the subarachnoid haemorrhage in the Silesian Province increased from 59.5 to 61.6 years in the analysed period. The most common group was patients aged 55–75 years. In the analysed population, the percentage of patients over 75 years increased, and the percentage of patients under 55 decreased. Research data show that the incidence of SAH increases with age, especially in females aged 55 and above. This trend is particularly evident in Japan in women over 75 [5,11,12]. The majority of patients hospitalized in the Silesian Province in 2009–2019 were women (58.2% of patients). This trend continued in each year of the follow-up. This is consistent with reports from other analyses, with some authors emphasizing that it depends on the age of patients, because in childhood and adolescence, bleeding from ruptured aneurysms occurs more often in males, while in the sixth decade of life the trend is reversed and such events occur twice as often in women [12]. The authors of a study conducted in Krakow also show that the majority of in-patients with non-ruptured aneurysms are women [13]. Our analysis showed that the rate of hypertension in patients hospitalized due to subarachnoid haemorrhage increased from 21% in 2009 to 39% in 2019. However, the observation period before SAH for the earlier years was much shorter than for the later years, and this is probably the main cause of the increasing trend for hypertension. While it was not possible to obtain data from other countries for the same time period, reports on changes in blood pressure in patients with SAH in 1980–2010 indicate a decreasing trend of systolic blood pressure and a slight increasing trend of diastolic blood pressure. World reports also showed a downward trend in smoking and its beneficial influence on the epidemiology of SAH [5]. 

The majority of the analysed patients were hospitalized in the neurosurgical ward (58.2%), and this rate decreased over the 10-year follow-up. Most likely this is related to the decreasing number of clipping of the ruptured aneurysm. The number of days of hospitalization also decreased. Almost half of all patients underwent intervention (coiling/other endovascular treatment or clipping), and such forms of treatment were undertaken more often in younger patients and in women. The number of interventions seems to be insufficient. There are major differences between hospitals in Silesia: the number of patients with SAH who underwent interventions was significantly larger in clinical and provincial hospitals (up to 73%) compared to municipal and district hospitals (in some less than 30%). We believe that the number of interventions was related mainly to diagnostic capabilities of a given hospital (the ability to diagnose an aneurysm), and availability and proximity of neurosurgery and interventional radiology units. In most medical facilities of Western Europe and USA, the number of patients with SAH who undergo intervention (coiling or clipping) is significantly larger (at least 60%) [14,15]. Similar findings can be found in older papers [12,16]. Although we excluded patients with diagnosed head trauma (classified in ICD-10 as S00-S09), it should be noted that it is not possible to clearly distinguish between atraumatic and traumatic SAH, and that the high number of untreated SAH patients included in the study could partially result from unrecognized traumatic SAH. 

Throughout the study period, 39.8% of patients underwent coiling or other endovascular treatment, while the remaining patients underwent clipping of the ruptured aneurysm. The number of patients undergoing endovascular treatment increased every year-from 28.7% in 2009 to 58.3% in 2019. It seems this is a countrywide trend. A survey conducted in 2019 among employees of 29 interventional units showed a slight advantage of the endovascular procedure over clipping [17]. It is believed that endovascular treatment of ruptured aneurysms is more beneficial than clipping, especially for older patients [18,19]. This benefit manifests itself in a greater percentage of patients without disability after coiling/other endovascular treatment compared to clipping [11]. Non-ruptured aneurysms are more and more often operated on in Japan, but this does not seem to reduce the number of SAH episodes [6]. 

SAH results in high mortality. In available reports, it varies between a quarter to one third of patients [1,2,7]. In our analysis, a slight upward trend in in-hospital mortality was observed. The case fatality of patients treated conservatively was more than two times higher than the mortality of patients who underwent intervention. In patients who suffered from SAH, the post-discharge 12-month mortality was also higher among those treated conservatively. This trend was especially true for elderly patients and was not affected by gender. This observation differs from the data available in publications from other countries. Worldwide data indicate a downward trend (up to 1.5% per year) in SAH mortality [12]. In Japan, despite the upward trend in the incidence, there is also a decrease in the number of fatal SAH, amounting for about 1% annually [7]. 

Among the analysed population, 4% of patients experienced recurrent SAH. Recurrent bleeding occurred both in the first and second half of the year after the primary bleeding, and also one year after the first incident. There are little data in the literature on recurrent SAH. Re-bleeding is observed mostly within the first 24 h after the primary episode [20]. The occurrence of cerebral amyloid angiopathy may contribute to recurrent SAH in the elderly [21]. The occurrence of recurrent subarachnoid haemorrhage is an adverse prognostic factor. The ISAT (International Subarachnoid Aneurysm Trial) study showed that the risk of rebleeding after surgery is low, but it is slightly more common in case of coiling. It was also found that rebleeding up to 5 years after the intervention occurred mainly after the endovascular procedure, while rebleeding more than 5 years after the intervention occurred mainly after the clipping procedure [19].

The presented analysis covers only one region in Poland. Unfortunately, apart from the subject of treatment methods for SAH, there are no publications concerning epidemiological data from all over the country. The available analyses come from individual cities (Krakow, Warsaw [1,8,13]) or centres such as the Internal Medicine Unit in Dabrowa Tarnowska [9] or the hospital in Krosno [10]. Those data were collected in much shorter time intervals (the longest follow-up was 3 years and concerned patients from Dabrowa Tarnowska) and included smaller groups of patients. Further analyses are needed to assess the epidemiological parameters of SAH nationwide.

## 5. Conclusions

Despite the number of stroke units increasing, in-hospital mortality in SAH patients is high, and the number of vascular interventions seems to be insufficient. Our findings call for better organization of care (diagnostic and therapeutic path) for SAH patients based in Poland. 

## 6. Limitations

This is an analysis of administrative, health insurer databases associated with reimbursement, not a prospective clinical registry. Unfortunately, we had no reliable information on previous episodes of SAH and therefore we were not able to report on the incidence. The only data available for analyses were hypertension risk factors (such as smoking), severity of SAH, functional status and pharmacological treatment. No additional clinical data (such as, for example, clinical history) were available. The amount of data needed to diagnose the cause of SAH were insufficient, thus the number of aneurysmal SAH remained unknown. We could only exclude patients with head trauma. During the follow-up observation, some patients could permanently migrate outside the Silesia Province and were lost to follow-up, however this effect should be marginal in populations after SAH. Further research involving patients from all over the country should be conducted.

## Figures and Tables

**Figure 1 jcm-11-04242-f001:**
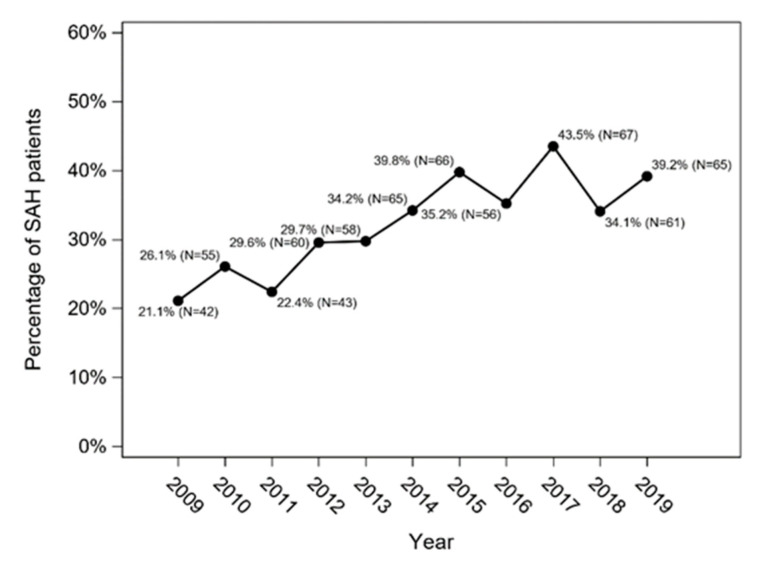
Percentage of SAH patients with previous history of hypertension in the Silesian Province between 2009 and 2019 (Cochran–Armitage trend test: *p* < 0.001).

**Figure 2 jcm-11-04242-f002:**
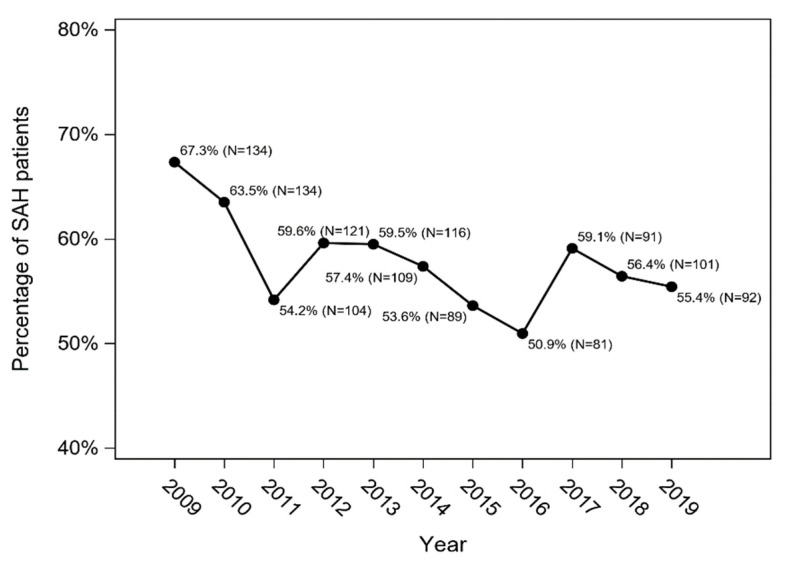
Percentage of SAH patients hospitalized in the neurosurgical departments of the Silesian Province between 2009 and 2019 (Cochran–Armitage trend test: *p* = 0.009).

**Figure 3 jcm-11-04242-f003:**
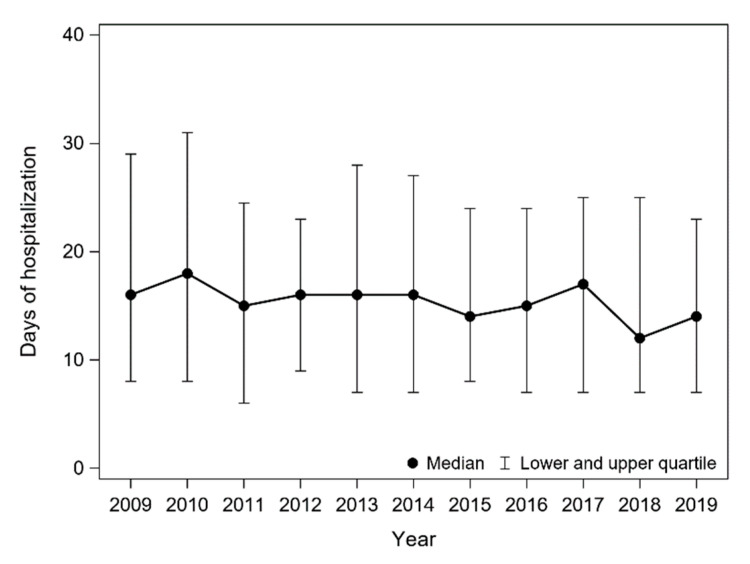
Median hospitalization time (from admission to the first ward to final discharge) for SAH patients in the Silesian Province between 2009 and 2019.

**Figure 4 jcm-11-04242-f004:**
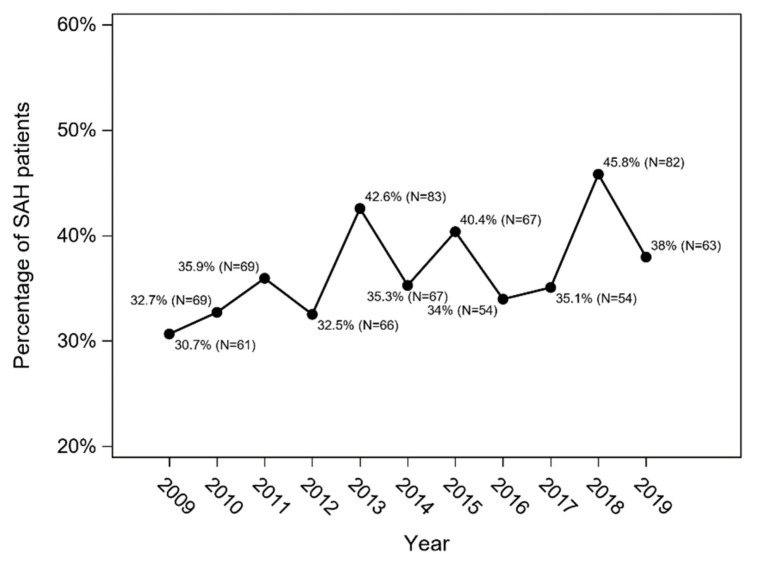
The overall in-hospital mortality in SAH patients in the Silesian Province between 2009 and 2019 (Cochran–Armitage trend test: *p* = 0.013).

**Figure 5 jcm-11-04242-f005:**
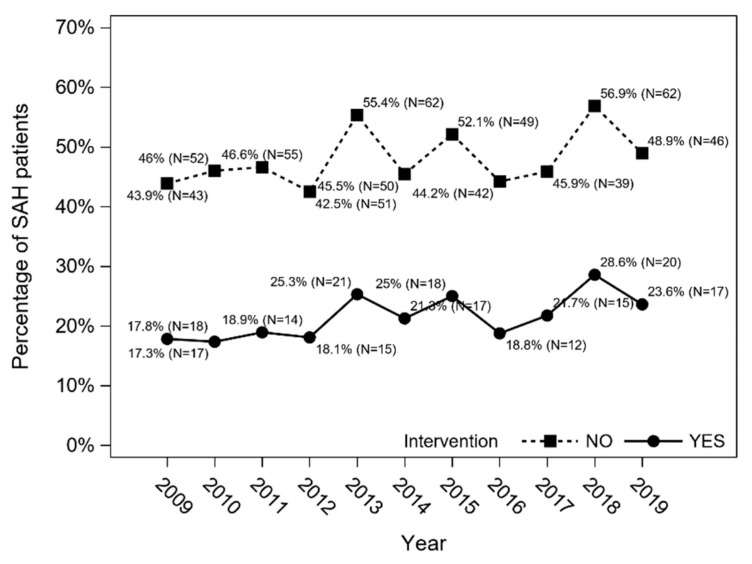
In-hospital mortality in SAH patients treated conservatively (no intervention) and in those who underwent intervention (Cochran–Armitage trend test with Holm–Bonferroni correction: *p* = 0.165 and *p* = 0.147, respectively).

**Figure 6 jcm-11-04242-f006:**
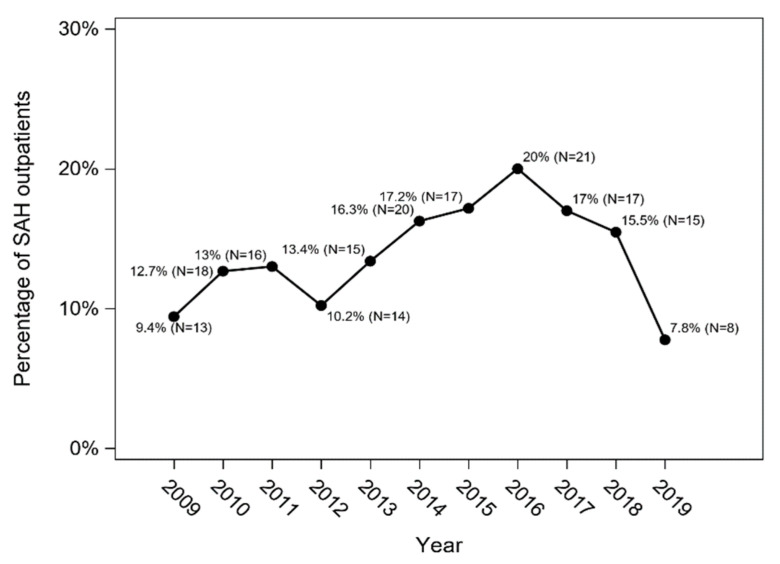
The overall post-discharge 12-month mortality in SAH patients (Cochran–Armitage trend test: *p* = 0.197).

**Figure 7 jcm-11-04242-f007:**
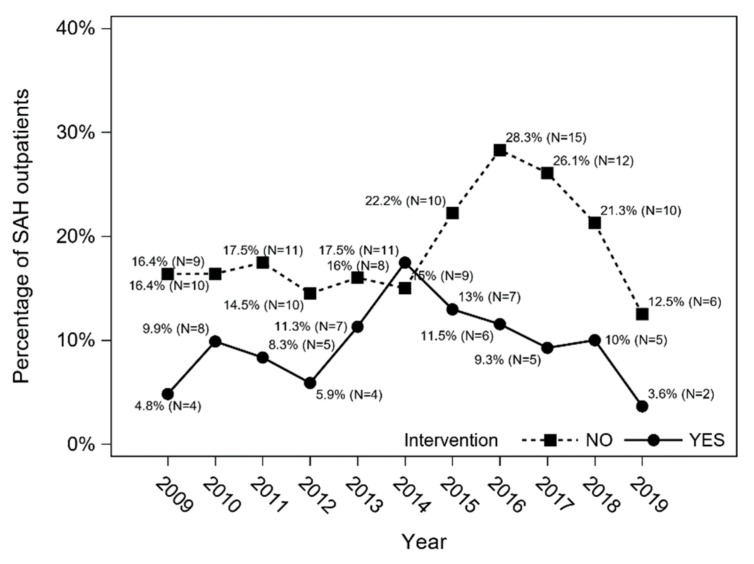
The post-discharge 12-month mortality in patients treated conservatively and in patients who underwent intervention (Cochran–Armitage trend test with Holm–Bonferroni correction: *p* = 0.529 and *p* = 0.617, respectively).

**Figure 8 jcm-11-04242-f008:**
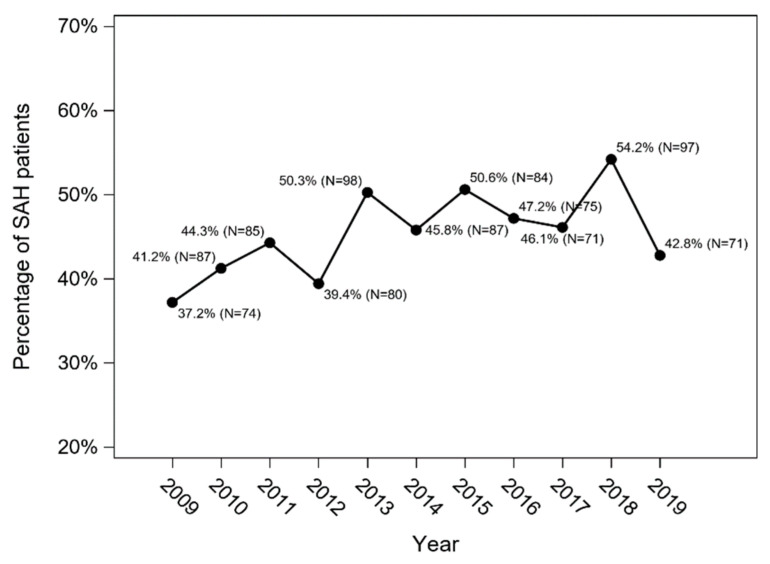
Overall combined mortality (in-hospital mortality and post-discharge 12-month mortality) (Cochran–Armitage trend test: *p* = 0.006).

**Figure 9 jcm-11-04242-f009:**
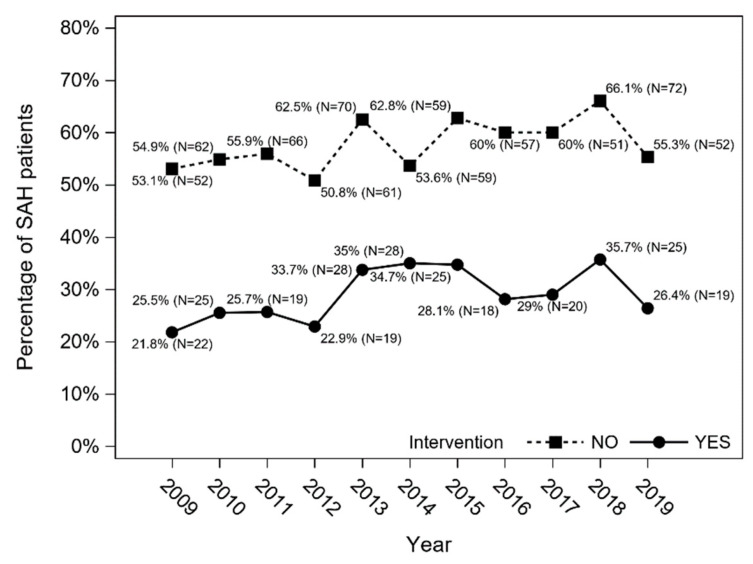
Combined mortality (in-hospital mortality and post-discharge 12-month mortality) in patients treated conservatively and in patients who underwent intervention (Cochran–Armitage trend test with Holm–Bonferroni correction: *p* = 0.152 and *p* = 0.152, respectively).

**Table 1 jcm-11-04242-t001:** The number of hospitalizations due to SAH (I60) in the Silesian Province between 2009 and 2019.

Year	Number of Hospitalizations Due to SAH N	Number of Hospitalizations per 100,000 Adult Inhabitants with 95% CI	Number of Hospitalizations in Women * N (%)	Number of Hospitalizations per 100,000 Adult Women with 95% CI	Number of Hospitalizations in Men * N (%)	Number of Hospitalizations per 100,000 Adult Men with 95% CI
Total	2014		1172 (58.2)		842 (41.8)	
2009	199	5.36 (4.64–6.16)	109 (54.77)	5.59 (4.59–6.74)	90 (45.23)	5.59 (4.59–6.74)
2010	211	5.67 (4.93–6.49)	121 (57.35)	6.21 (5.15–7.42)	90 (42.65)	6.21 (5.15–7.42)
2011	192	5.15 (4.45–5.93)	114 (59.38)	5.84 (4.81–7.01)	78 (40.63)	5.84 (4.81–7.01)
2012	203	5.44 (4.72–6.25)	110 (54.19)	5.63 (4.63–6.78)	93 (45.81)	5.63 (4.63–6.78)
2013	195	5.24 (4.53–6.03)	111 (56.92)	5.69 (4.68–6.85)	84 (43.08)	5.69 (4.68–6.85)
2014	190	5.11 (4.41–5.89)	122 (64.21)	6.25 (5.19–7.47)	68 (35.79)	6.25 (5.19–7.47)
2015	166	4.47 (3.82–5.21)	101 (60.84)	5.19 (4.23–6.30)	65 (39.16)	5.19 (4.23–6.30)
2016	159	4.29 (3.65–5.01)	97 (61.01)	4.99 (4.05–6.09)	62 (38.99)	4.99 (4.05–6.09)
2017	154	4.17 (3.54–4.88)	97 (62.99)	5.00 (4.06–6.10)	57 (37.01)	5.00 (4.06–6.10)
2018	179	4.86 (4.18–5.63)	89 (49.72)	4.61 (3.70–5.67)	90 (50.28)	4.61 (3.70–5.67)
2019	166	4.53 (3.87–5.27)	101 (60.84)	5.25 (4.28–6.38)	65 (39.16)	5.25 (4.28–6.38)
*p*		0.0005 *	NS *	0.026 **	NS *	0.010 **

* Cochran–Armitage trend test. ** Cochran–Armitage trend test with Holm–Bonferroni correction. The 95% confidence intervals (CIs) were calculated assuming the gamma distribution.

**Table 2 jcm-11-04242-t002:** The number of hospitalizations due to SAH according to age category in the Silesian Province between 2009 and 2019.

Year	Age Mean ± SD	Age < 55 Years N (%)	Age ≥ 55 and ≤75 Years N (%)	Age > 75 Years N (%)
Total	62.2 ± 14.4	637 (31.6)	945 (46.9)	432 (21.5)
2009	59.5 ± 14.2	79 (39.70)	87 (43.72)	33 (16.58)
2010	60.6 ± 13.9	77 (36.49)	101 (47.87)	33 (15.64)
2011	61.5 ± 13.4	62 (32.29)	96 (50.00)	34 (17.71)
2012	60.9 ± 14.9	72 (35.47)	91 (44.83)	40 (19.7)
2013	62.1 ± 13.5	60 (30.77)	99 (50.77)	36 (18.46)
2014	61.6 ± 13.8	58 (30.53)	92 (48.42)	40 (21.05)
2015	65.6 ± 14.4	40 (24.10)	76 (45.78)	50 (30.12)
2016	63.9 ± 14.8	40 (25.16)	79 (49.69)	40 (25.16)
2017	63.8 ± 15.2	43 (27.92)	68 (44.16)	43 (27.92)
2018	64.3 ± 14.8	49 (27.37)	82 (45.81)	48 (26.82)
2019	61.6 ± 15.4	57 (34.34)	74 (44.58)	35 (21.08)
*p*	<0.001 *	0.006 **	0.712 **	<0.001 **

* Jonckheere–Terpstra test. ** Cochran–Armitage trend test with Holm–Bonferroni correction.

**Table 3 jcm-11-04242-t003:** The number of interventions in patients with SAH in the Silesian Province.

Year	All Interventions (Coiling/Other Endovascular Treatment or Clipping of the Ruptured Aneurysm) N (% of All SAH Patients)	Coiling/Other Endovascular Treatment of the Ruptured Aneurysm N (% of All Interventions)
Total	866 (43%)	345 (39.8%)
2009	101 (50.7%)	29 (28.7%)
2010	98 (46.4%)	35 (35.7%)
2011	74 (38.5%)	30 (40.5%)
2012	83 (40.9%)	30 (36.1%)
2013	83 (42.6%)	23 (27.7%)
2014	80 (42.1%)	32 (40.0%)
2015	72 (43.4%)	26 (36.1%)
2016	64 (40.3%)	27 (42.2%)
2017	69 (44.8%)	41 (59.4%)
2018	70 (39.1%)	30 (42.9%)
2019	72 (43.4%)	42 (58.3%)
*p*	0.1734 *	<0.001 *

* Cochrane–Armitage trend test.

## Data Availability

The data presented in this study is available on request from the corresponding author. The data is not publicly available due to privacy reasons.

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
