# Peer review of "Subarachnoid Haemorrhage—Incidence of Hospitalization, Management and Case Fatality Rate—In the Silesian Province, Poland, in the Years 2009–2019"

_jcm, 2022, doi:10.3390/jcm11144242_

Round 1

Reviewer 1 Report

The manuscript is well written and clinically sound. However, in the introduction section, some references should be added:

1) Subarachnoid hemorrhage (SAH) is a type of haemorrhagic stroke that is  characterized by high morbidity and fatality. (REFERENCE) 

2) After excluding traumatic and iatrogenic causes, the most common reason of SAH is ruptured cerebral aneurysm. (REFERENCE) 

3) Arteriovenous malformation, cavernous haemangioma, cerebral arteriovenous fistula, and venous thrombosis could also be the reason of blood extravasation. (REFERENCE). 

4) Other rarer causes include tumour bleeding, vasculitis, encephalitis, cerebral amyloid angiopathy, postpartum eclampsia and reversible cerebral vasoconstriction syndrome (RCVS). (REFERENCE)

Moreover, some mistakes must be corrected (for example:

-reason for (not of)

-administrative databases   (pluaral),

-An increase in  in-hospital  mortality (not of), sequalae not sequalae

-incidence of SAH decreased by about 40% in Europe, 46% in Asia, and 14% (not decrease)

Author Response

Thank you very much for review of our paper and all the valuable remarks which make our work much better.

Responses to particular remarks:

Point 1

„The manuscript is well written and clinically sound. However, in the introduction section, some references should be added:

1) Subarachnoid hemorrhage (SAH) is a type of haemorrhagic stroke that is  characterized by high morbidity and fatality. (REFERENCE) 

2) After excluding traumatic and iatrogenic causes, the most common reason of SAH is ruptured cerebral aneurysm. (REFERENCE) 

3) Arteriovenous malformation, cavernous haemangioma, cerebral arteriovenous fistula, and venous thrombosis could also be the reason of blood extravasation. (REFERENCE). 

4) Other rarer causes include tumour bleeding, vasculitis, encephalitis, cerebral amyloid angiopathy, postpartum eclampsia and reversible cerebral vasoconstriction syndrome (RCVS). (REFERENCE)”

Response 1:

I have added the references in the introduction according to the review:

Subarachnoid haemorrhage (SAH) is a type of haemorrhagic stroke characterized by high morbidity and mortality [1,2]. After excluding traumatic and iatrogenic causes, the most common reason for SAH is ruptured cerebral aneurysm [2]. Arteriovenous malformation, cavernous haemangioma, cerebral arteriovenous fistula, and venous thrombosis could also be the reason for blood extravasation [3]. Other rarer causes include tumor bleeding, vasculitis, encephalitis, cerebral amyloid angiopathy, postpartum eclampsia and reversible cerebral vasoconstriction syndrome (RCVS) [3,4].

Point 2:

„Moreover, some mistakes must be corrected (for example:

-reason for (not of)

-administrative databases   (pluaral),

-An increase in  in-hospital  mortality (not of), sequalae not sequalae

-incidence of SAH decreased by about 40% in Europe, 46% in Asia, and 14% (not decrease)”

Response 2:

I have corrected these mistake, and beside I make English correction of the whole manuscript.

All changes (according to the 1. and 2. Review) are in red.

Thank you once again for valuable review.

With regards

Reviewer 2 Report

Thank you very much for the opportunity to review the manuscript jcm-1763109 entitled "Subarachnoid haemorrhage – hospitalized incidence, management and case fatality - in the Silesian Province, Poland, during the decade 2009-2019". 

The authors managed to analyze a large regional population with subarachnoid hemorrhage. They must have spent a lot of effort collecting and analyzing the data. Interestingly, the authors detect a large number of untreated SAH patients. Unfortunately, there is a significant weakness in this study. The high rate of untreated patients remains mainly unexplained; e.g., p.4 l.113ff "Interventions (coiling/other endovascular treatment or clipping of the ruptured aneurysm) were performed in 866 cases (43%) (Table 3). The remaining patients were treated conservatively (n=1148; 57%)."

·      This subgroup needs to be better analyzed (cause of SAH, diagnostics, health state at admission)

·      A better discussion of why this group of untreated SAH patients is this high  

·      Please explain how did you exclude atraumatic SAH?

·      https://doi.org/10.1161/01.STR.31.5.1054

The study's scientific value is highly questionable if the authors cannot sufficiently explain this high number of untreated "atraumatic" SAH patients.  I know that with the retrospective study design is hard to get the data, but the design is also prone to bias. 

Author Response

Thank you very much for review of our paper and all the valuable remarks.

Point 1:

„Unfortunately, there is a significant weakness in this study. The high rate of untreated patients remains mainly unexplained; e.g., p.4 l.113ff "Interventions (coiling/other endovascular treatment or clipping of the ruptured aneurysm) were performed in 866 cases (43%) (Table 3). The remaining patients were treated conservatively (n=1148; 57%). This subgroup needs to be better analyzed (cause of SAH, diagnostics, health state at admission). A better discussion of why this group of untreated SAH patients is this high.”

Response 1:

The numer of untreated SAH patients was also a surprise for us. But the statistics were performed reliably. We used the database of National Health Fund, where all procedures (including coiling and clipping) are compulsorily coded. When preparing the article after the review, we checked the results and obtained the same calculations.

We try to characterize both groups:

“Subjects who underwent intervention were younger (56.8 ± 7 vs 67.1 ± 11 years), more often female (63.4% vs 54.3%) and less frequently diagnosed with hypertension (21% vs 40%) than those treated conservatively.”

Other clinical data were unavailable (we discribed it in the Limitation):

“No additional clinical data was available for the analyses (like for example clinical history), other than hypertension risk factors (like smoking), severity of SAH, functional status or pharmacological treatment. We have not enough data to establish the cause of SAH, so the number of aneurysmal SAH was unknown. We were only able to exclude patients with head trauma.”

We also tried to explain such a small number of interventions in the discussion:

“The number of interventions seems to be insufficient. There are very large differences between hospitals in Silesia. The number of patients with SAH who underwent interventions was much greater in clinical and provincial hospitals (up to 73%) than in municipal and district hospitals (in some less than 30%). In our opinion the number of interventions was mainly related to diagnostic possibilities in the hospital (the ability to detect an aneurysm), availability and proximity of neurosurgery and interventional radiology. In most centers of Western Europe and USA, the number of patients with SAH undergoing intervention (coiling or clipping) is much greater (60% or more). Similar data can be found in some older papers [12,16]. It means that better organization for care of SAH patients is needed in Poland.”

Point 2:

Please explain how did you exclude atraumatic SAH?

Response 2:

In Material and Methods we explain the used definition of atraumatic SAH:

“it means that subjects with comorbid diagnosis: head trauma (classified in ICD-10 as S00-S09) were excluded”

Also I make English correction of the whole manuscript.

All changes (according to the 1. and 2. Review) are in red.

Thank you once again for valuable review.

With regards

Round 2

Reviewer 2 Report

As the main limitation cannot be improved by the data. I do not see any value. Atraumatic and traumatic SAH cannot be clearly distinguished and the high number of untreated SAH is most likely caused by traumatic SAH cases. 

Author Response

Reviewer 2:

Thank you very much for further review of our paper and all the valuable remarks.

Point 1:

As the main limitation cannot be improved by the data. I do not see any value. Atraumatic and traumatic SAH cannot be clearly distinguished and the high number of untreated SAH is most likely caused by traumatic SAH cases. 

Response 1:

We added a sentence in the discussion:

„Although we have excluded patients with diagnosed head trauma (classified in ICD-10 as S00-S09), it should be noted that it is not possible to clearly distinguish between atraumatic and traumatic SAH, and that the high number of untreated SAH patients included in the study could partially result from unrecognized traumatic SAH.” 

Also we modify conclusion:

“Our findings call for better organization of care (diagnostic and therapeutic path) for SAH patients  based in Poland. “

We also checked all the cited references and make further English correction of the whole manuscript.

New changes are given in green.

Thank you once again for valuable review.

With regards